# Spotlight on the Multiscale Structural and Physicochemical Properties of Red Adzuki Bean Starch through Partial Amylose Removal Combined with Hydrochloric Acid

**DOI:** 10.3390/foods12183366

**Published:** 2023-09-07

**Authors:** Xinyue Liu, Zhuangzhuang Sun, Wenqing Zhao, Jiayu Zheng, Wei Liang, Wenhao Li

**Affiliations:** Shaanxi Union Research Center of University and Enterprise for Grain Processing Technologies, College of Food Science and Engineering, Northwest A&F University, Xianyang 712100, China; liuxinyue0615@163.com (X.L.); sunzhuangz@nwafu.edn.cn (Z.S.); zhaowq2021@163.com (W.Z.); jiayu.zheng@nwafu.cn (J.Z.); liangwei@nwafu.edu.cn (W.L.)

**Keywords:** deamylose, dual modification, red adzuki bean starch, multiscale structure

## Abstract

To explore the effect of amylose within starch granules on the efficiency of starch hydrolysis by acid, we chose the warm water extraction method to treat red adzuki bean starch to obtain different degrees of amylose removal granule models and to prepare samples in combination with acid hydrolysis. The amylose content was reduced after acid hydrolysis, reducing the peak viscosity (2599–1049 cP), while the solubility was significantly increased. In contrast, the short-chain content of the deamylose–acid hydrolysis samples was reduced considerably, exacerbating the trend towards reduced starch orderliness and increased solubility. This work reveals the granular structure of starch from the point of view of deamylose and contributes to a thorough understanding of the mechanisms of acid hydrolysis. It might add to knowledge in starch science research and industrial applications for the acid processing of starch-based foods, particularly with regard to the most important factors controlling the structure and function of starch.

## 1. Introduction

Starch is an important renewable resource in nature and is widely used in food, medical, biological and material applications [1]. However, native starch has low solubility and poor properties. There are many problems in the application of starch. This is due to its large molecular configuration, which makes it poorly mobile, strong intermolecular hydrogen bonding, high molecular activation energy, easy water absorption, which increases aging and spoilage, insolubility in cold water, poor shear resistance, thickening after heating and poor thermal stability [2], and therefore its large-scale application in industry is extremely limited.

Starch consists of two types of dextran: amylose and amylopectin. Amylose is a relatively long and poorly branched polymer, whereas amylopectin is a highly branched polymer. The starch chains are linked by α-(1,4) bonds and branched by α-(1,6) bonds. Amylopectin side chains can form continuous amorphous and crystalline flakes [3]. The intrinsic structure of starch is complex and is thought to be organized at different levels: granular, laminar, crystalline, short-range ordered, helical and molecular [4]. Differences in the structural characteristics of these starches can often determine the quality of the food products in which they are the main ingredient. The molecular structure of amylose is dense and highly ordered. Certain modifications do not easily change these disadvantageous properties [5,6]. Low-amylose rice could be utilized to develop extrudates with better physicochemical properties [6]. Borah et al. [7] reported honeycomb structure and better pasting properties for extruded weaning food formulation developed using low-amylose rice. Therefore, the plan of this research is to remove the high amylose content of red adzuki bean starch by using warm water modification to make it more amenable [2].

Moreover, acid hydrolysis has also been proven to produce low-molecular-weight starches with a large reduction in liquid viscosity after paste formation, which has led to more and better applications in the food industry, such as fondant and starch jelly, in the paper industry as a surface sizing agent, to improve printability, and in the textile industry for warp sizing [8,9]. High-amylose starch always exhibits high resistance to acid hydrolysis. In contrast, low-amylose starch is highly sensitive to acid hydrolysis, likely due to the tightness of the double helix within the starch crystals [10]. In recent studies, the particle and molecular structure of acid-hydrolyzed maize starches with different amylose contents and their corresponding physicochemical properties have been investigated in detail. It was found that both acid concentration and hydrolysis time systematically reduced the molecular weight of the starch and the chain length of the amylose and amylopectin fractions. Furthermore, acid dilution slightly reduced the stability of high-amylose maize starch gels, but did not change the stability of waxy maize starch gels [11,12]. 

Much effort has been put into further understanding the structure and properties of different starches in terms of amylose content to obtain better industrial properties through acid hydrolysis. However, the presence of amylose has prevented a clear understanding of the degradation characteristics of starches in terms of structure and properties [13]. We hypothesized that removing some amylose could maximize this advantage of acidolytically modified starch properties. In this study, the multi-level structural and physicochemical properties of red adzuki bean starch with a high amylose content were investigated by using a deliquescence treatment and different concentrations of hydrochloric acid, and the effects of acid hydrolysis on the “structure-property” relationship were also elucidated by first removing the amylose.

## 2. Materials and Methods

### 2.1. Material

Red adzuki beans were bought from Yangling Haoyouduo supermarket (Xianyang China). 8-aminopyrene-1,3,6-trisulfonate sodium salt (APTS) was obtained from Aladdin Reagent Co., Ltd., Shanghai, China. All other chemical reagents were of analytical grade.

### 2.2. Starch Separation

The red adzuki bean starch was isolated by using the method of Li et al. [14] with minor alterations. The wet milling method was used: soaking the beans in distilled water until softened (25 °C, 24 h), then homogenizing the beans with a pulper. The pulp was sieved through a 150 µm nylon sieve, and the sieved extract was allowed to stand until the starch precipitated. The starch sediment was then resuspended twice in distilled water. Afterwards, the starch suspension was centrifuged (1780× *g*, 15 min), and the supernatant and the top cream-colored layer were discarded. The purified starch isolate was dried in a hot air oven at 45 °C for 1 d.

### 2.3. Deamylose Treatment

Then, 30 g red adzuki bean starch was dispersed in 300 mL methanol (85%, *v*/*v*; 3000 rpm, 24 h). The residue was air-dried in a fume hood after standing for 1 h to obtain defatted starch. Next, 10 g of the above starch was mixed with distilled water (300 mL) and placed in a water bath at 50 °C for 1 h and 2.5 h. The supernatant was removed (4000 rpm, 1 h), washed twice with distilled water and centrifuged (2325× *g*, 5 min). Finally, the residue was freeze-dried and passed through a 150 µm sieve. The samples were named DA-1 and DA-2.5.

### 2.4. Hydrochloric Acid Modification Treatment

First, 10% starch emulsion was added to HCl solutions at concentrations of 0.3 mol/L, 0.6 mol/L and 0.9 mol/L and reacted for 2 h in a constant-temperature shaker at 30 °C, and then the reaction was terminated with NaOH (0.5 M). The sample was washed with distilled water and the process was repeated thrice (2325× *g* for 5 min). The residue was placed in a blast-drying oven at 40 °C, crushed and passed through a 150 µm sieve. The samples obtained were named H-0.3, H-0.6 and H-0.9.

### 2.5. DA—HCl Treatment

The samples obtained from the process described in Section 2.3 were processed using the method in Section 2.4 to prepare the composite treated samples. The resulting samples were DA1-H0.3, DA1-H0.6, DA1-H0.9, DA2.5-H0.3, DA2.5-H0.6 and DA2.5-H0.9.

### 2.6. Microscopic Inspection

#### 2.6.1. Scanning Electron Microscopy (SEM)

Starch samples were attached to an aluminum loading platform with double-sided conductive tape, and the starch was gilded. Then, samples were taken out, observed and photographed with a Nano SEM-450 (FEI, Hillsboro, OR, USA) at 5.0 kV and 3000× magnification.

#### 2.6.2. Confocal Laser Scanning Microscopy (CLSM) 

First, 2 mg starch was dispersed in a mixture of 4 µL APTS (10 mM) and 4 µL sodium cyanoborohydride (1 M) (30 °C, 15 h). The reacted sample was washed with distilled water and then dispersed in 20 µL of a glycerol–water mixture (50%, *v*/*v*). Finally, a drop was taken on a slide for observation using a CLSM (LEICA TCSSP8, Wetzlar, Germany) fitted with a 100× plan apo/1.4011 oil UV. 

#### 2.6.3. Polarizing Light Microscope (PLM) 

A small amount of the sample was homogenized with water–glycerol (1:1, *v*/*v*), and the sample morphology was observed using light microscopy under a 400× objective.

### 2.7. Chain Length Distribution (CLD) Measuring

The methodology of CLD was established by the program presented by Su et al. [15], which was monitored using a high-performance anion exchange chromatography pulsed amperometric detector ICS-5000+ (Thermo Fisher Scientific, Pleasanton, CA, USA). The analytical column was a Dionex CarboPac™ PA-100. 

Sample preparation: starch (40 mg) was dispersed in 2 mL acetate buffer (0.01 M, pH 4.5) and pasted in a boiling water bath for 10 min. After cooling to 25 °C, 5 µL pullulanase was added and reacted (50 °C, 24 h). The enzyme was deactivated in a boiling water bath for 20 min, cooled and centrifuged (2325× *g*, 10 min), and the supernatant was diluted 50-fold and passed through a 0.22 μm aqueous membrane in a sample bottle. 

### 2.8. X-ray Diffraction (XRD) Measurement

The crystalline structure of the samples was determined using an X-ray diffractometer (Rigaku Corporation, Tokyo, Japan). The condition parameters were as follows: Cu-Kα radiation; 2θ: 4° to 60°; scanning rate: 6°/min. 

### 2.9. Short-Range Ordering

Fourier transform infrared (FT-IR) red adzuki bean starch spectra were collected with a Vertex 70 spectrometer; scanning range: 4000 to 400 cm^−1^, scanning numbers: 16, resolution was 4 cm^−1^.

### 2.10. Amylose Content

The amylose content of the red adzuki bean starch was determined by referring to an iodine-binding procedure of Gao et al. [16]. First, 50 mg starch was mixed with 0.5 mL ethanol and 4.5 mL NaOH (1 M), then boiled in a water bath for 10 min and cooled. Then, 25 mL petroleum ether was added (3000 rpm, 10 min). After 15 min, the upper layer of petroleum ether was aspirated and placed in an oven at 60 °C to evaporate completely. The absorbance at 620 nm was determined by reacting 2.5 mL of the sample solution with potassium iodide.

### 2.11. Measurement of Pasting Properties

An RVA-Tec Master (Newport Scientific Co., Ltd., Sydney, Australia) is the device used for pasting property measurement. First, 3.0 g starch and 25 mL deionized water were mixed, which was calculated to a wet basis of 14%. Red adzuki bean starch was equilibrated at 50 °C for 1 min, and then was heated at a rate of 12 °C/min to 95 °C, held for 2.5 min, then cooled to 50 °C at the same speed and held for 2 min. 

### 2.12. Measurement of Solubility and Swelling Power

About 0.5 g starch (m_1_) was dispersed in 25 mL distilled water, and the prepared 2% starch solution was placed in a centrifuge tube. The tubes were centrifuged after shaking for 0.5 h in a water bath at different temperatures (50 °C, 60 °C, 70 °C, 80 °C and 90 °C). The supernatant was then poured into an aluminum box and dried, and the box (m_2_) and the centrifuge tube (m_3_) were weighed. The solubility and swelling power were determined as follows:Solubility = m_2_/m_1_ × 100(1)
Swelling power = m_3_/m_1_ × (100 − S)(2)

### 2.13. Statistics Analysis

All statistical calculations were performed in SPSS version 21.0. Analysis was executed by using ANOVA, and Duncan’s test (*p* < 0.05) was applied for multiple comparisons. Principal component analysis (PCA) and Pearson’s correlation analysis were analyzed using Origin 2023.

## 3. Results

### 3.1. Morphological Characteristics

Observation of the microscopic images of red adzuki bean starch (Figure 1) showed that the starch granules remained intact after a single treatment, and the DA-hydrochloric acid treatment destroyed the integrity of the starch granules and increased the breakage rate with the increase in the treatment intensity. There was no significant difference in the growth ring structure and polarization properties of all the samples.

### 3.2. Chain Length Distribution

The results of the chain length distribution of branched starch before and after modification of red adzuki bean starch are presented in Table 1. Compared to native starch, warm water removal of amylose or acid hydrolysis alone decreased the proportions of A and B1 chains and increased the proportions of B2 and B3 chains. There were no significant differences between DA-hydrochloric acid treatments and single treatments, with higher concentrations of hydrochloric acid further decreasing the proportions of the A chains for the same warm water treatment time.

### 3.3. X-ray Diffraction (XRD)

As can be seen from Figure 2A, no treatments changed the C-type crystalline structure of red adzuki bean starch. However, with the prolongation of the warm water treatment time, or the increase in the concentration of hydrochloric acid modification treatment, the crystallinity of red adzuki bean starch first increased and then decreased. In addition, DA-HCl treatment was able to significantly disrupt its crystalline structure, further reducing the crystallinity of the single-modified sample.

### 3.4. FT-IR Spectra

FT-IR was used to detect the degree of order in the short-range ordered molecular structure of starch. The FTIR spectra of modified starch were similar to those of native starch, and no new absorption peaks were generated (Figure 2B). With the increase in hydrochloric acid addition, the 1047 cm^−1^/1022 cm^−1^ ratio tended to increase and decrease. The removal of amylose using warm water increased the ratio. In addition, DA-HCl treatment resulted in a gradual but slight decrease in the short-range ordering of red adzuki bean starch.

### 3.5. Amylose Content

As shown in Table 1, the amylose content of red adzuki bean starch decreased with hydrochloric acid concentration and the amylose content further decreased with the prolongation of the time of removing amylose with warm water. The combined treatment exacerbated this phenomenon.

### 3.6. Pasting Properties

Table 2 analyzes the pasting characteristics of red adzuki bean starch pre- and post-modification. The peak, trough and final viscosities decreased with the warm water removal time. However, the attenuation value, regeneration value, gelatinization temperature and pasting time were gradually enhanced. In addition, the peak viscosity, final viscosity and trough viscosity of red adzuki bean starch also decreased after hydrochloric acid treatment and were negatively correlated with hydrochloric acid concentration. The changes in pasting time and pasting temperature were not significant. The pasting characteristics of DA-HCl samples were not significantly different from those of the individually treated samples.

### 3.7. Solubility and Swelling Power

Table 3 reveals the solubility and swelling power of native, DA, hydrochloric-acid-modified and dual-modified red adzuki bean starch.

At 50–70 °C, the solubility of the samples increased slightly with increasing hydrochloric acid concentration. At higher temperatures, the solubility of the samples first decreased and then increased with hydrochloric acid concentration, and the swelling power of the starch granules decreased. However, the solubility and swelling power of DA starch increased gradually with increasing temperature. However, the solubility and swelling power of the war- water-modified sample were much lower than that of native starch.

## 4. Discussion

### 4.1. Morphological Characteristics

Microscopic images of the native and deamylose- and hydrochloric-acid-treated red adzuki bean starch granules are shown in Figure 1. The SEM showed that most native red adzuki bean starch granules had smooth oval or kidney-shaped surfaces. At the same time, a few had raised surfaces with prominent internal grooves or even cracks. The surface of the starch granules became depressed after the extraction of the amylose with warm water, and the percentage of deformation increased with the increasing time of deamylose. However, the granules still maintained their intact form. 

In contrast, adding low concentrations of HCl did not significantly affect the red adzuki bean starch granules, which had a smooth surface without cracks and depressions. As the HCl concentration increased, wrinkles appeared on the granule surface, possibly caused by HCl corroding the starch granules. Similar results were obtained for treating pea starch with hydrochloric acid [17]. In general, none of the single modification treatments destroyed the integrity of the starch granules. In contrast, the DA-HCl treatment disrupted their integrity and increased the breakage rate with longer deamylose times and higher HCl concentrations. This may be because the deamylose process interfered with the crystal structure of the amorphous region and the dissociation of the double helix structure, which destroyed the amorphous region of the starch [18]. At this point, the pre-deamylose treatment made it easier for HCl to enter the red adzuki bean starch granules, increasing corrosion damage.

In the CLSM diagram, it is possible to observe the clear growth rings and umbilical dot structure in the native red adzuki bean starch (Figure 1(B1)). There were no significant differences in the growth rings and lightness of the samples before and after the HCl modification. Nevertheless, the starch granules underwent deformation as the warm water deamylose time increased. Moreover, the structure of the starch growth rings could be observed after DA-HCl treatment, but the proportion of starch particle deformation was further increased. According to the optical microscope results in Figure 1(C1)–(C12), the native, DA, HCl-treated and dual-modified red adzuki bean starch showed a polarized cross, indicating that none of the treatments affected the polarized characteristics. This was also consistent with the obtained CLSM results.

### 4.2. Chain Length Distribution

After a short time of deamylose in warm water, it was found that the proportion of the A chain and B1 chain were significantly lower than that of the native starch, and the proportion of the B2 chain and B3 chain increased. With the extension of treatment time, the proportion of the A chain reduced and the proportion of the B3 chain increased, but the difference was insignificant between DA-1 and DA-2.5. This phenomenon indicated that a short period of warm water deamylose treatment caused the amylose with a smaller molecular weight on the surface of the starch to move out. As the water penetrated the granules, it broke the intermolecular glycosidic bonds of the starch and these underwent breaking and rearrangement [19], decreasing the A and B1 chains; the proportion of B2 and B3 chains increased. However, as the time for amylose removal using warm water increased, the structure of the crystalline region was disrupted, and rearrangements occurred, resulting in a further increase in the proportion of B2 and B3 chains. This conclusion was shared with the CLSM images, where the fluorescence intensity at the umbilical point position of the red adzuki bean starch granules was significantly reduced.

Hydrochloric acid first destroyed the amorphous region of the starch, and then the crystalline region [20]. The contact area between hydrochloric acid and starch particles increased with the release of short chain molecules with small molecular weight, thus destroying the crystalline structure of starch granules. As the concentration of hydrochloric acid increased gradually, the crystalline zone was further disrupted. There was no significant difference between 0.6% and 0.9% HCl addition, indicating that the acid hydrolysis degree of red adzuki bean starch granules had reached its best with a low concentration of hydrochloric acid, and a higher concentration of hydrochloric acid was required for the crystallization zone of the starch granules. DA-HCl treatment revealed that the effect of HCl on starch granules was less for the same warm water treatment time and a higher concentration of HCl would further reduce the proportion of A chains. The results demonstrated that the crystal structure of red adzuki bean starch was more stable after water bath deamylose treatment [21].

### 4.3. X-ray Diffraction (XRD) Interpretation

Figure 2A shows that red adzuki bean starch has strong peaks at 5.5°, 15°, 17° and 23°, typical of C-type starch. There was no significant difference between the water content of treated starch and that of native starch. It was found that neither amylose nor acid hydrolysis could alter the crystal type of starch. Although the shape of the diffraction peak was not changed, it is noteworthy that the crystallinity increased and then decreased with increasing time of warm water deamylose treatment (Table 1). This result may be because amylose may reduce the ordering of crystallites and amylopectin double helices by disrupting their alignment, preferred orientation and packing [4].

In addition, as the concentration of HCl modification treatment increased, the crystallinity of red adzuki bean starch increased and then decreased, with the highest at HCl-0.3. This short period of mild acid hydrolysis mainly affects the amorphous regions of the starch, resulting in a finer crystallinity of the residual starch [22]. This result also corroborated the previous findings [17]. However, the crystalline region of red adzuki bean starch was further acidified with increasing HCl concentration, which in turn reduced the crystallinity. The DA-HCl treatment sample was found to be significantly less crystalline than that of the single modified treatment, indicating that DA-HCl treatment was able to disrupt the crystalline structure of the red bean starch.

### 4.4. FT-IR Spectra

Figure 2B presents the FT-IR spectra of red adzuki bean starch in the wavenumber range of 400–4000 cm^−1^. The FT-IR spectra of the modified starch are similar to those of the native starch, with no new absorption peaks generated, indicating that the modified treatments created no new functional groups or chemical bonds. The absorbance value calculated at 1047 cm^−1^/1022 cm^−1^ could reflect the degree of order in the proximally ordered structure of the starch (Table 1). The higher the value of 1047/1022, the more ordered the starch molecules are [4].

The 1047 cm^−1^/1022 cm^−1^ ratio exhibited a trend of increasing and then decreasing with the increase in hydrochloric acid addition. This indicates that the orderliness of red adzuki bean starch increased and then decreased after hydrochloric acid treatment. This was also consistent with the crystallinity results. The low concentration of hydrochloric acid first acidifies the amorphous regions of the red adzuki bean starch granules, the broken molecular fragments rearrange themselves and the free amylose chains caused by acid cleavage of some of the amylose molecules might contribute to the formation of new crystal structure, leading to an increase in short-range ordering [22]. However, as the concentration of hydrochloric acid increased, the crystalline region was also disrupted, increasing the short-range order. The ratio of 1047 cm^−1^/1022 cm^−1^ gradually increased with the increase in the deamylose treatment time in a water bath, which might be because the hydrogen bonding force between molecules was weakened after the expansion of the starch particles, and the short-range order increased when amylose was permeated. This result corroborated the previous findings [19].

Moreover, the DA-HCl treatment resulted in a gradual but slight change in the short-range orderliness of the red adzuki bean starch. It demonstrated hydrochloric acid was less acidolytic to the sample after the water bath deamylose. This may be due to the rearrangement of starch molecular chains to form a more stable crystalline structure after the red adzuki bean starch deamylose treatment [21]. It was also possible that the reduction in amylose content, i.e., the decline in the single helix structure of the hydrochloric acid digestion and the degree of acid digestion [22], resulted in a lower but insignificant change in the 1047 cm^−1^/1022 cm^−1^ ratio.

### 4.5. Amylose Content Analysis

It is evident from Table 1 that the amylose content decreased significantly with increasing hydrochloric acid concentration. This might be because acidolysis mainly acted on the amorphous region of the red adzuki bean starch granules, destroying the glucoside bond of amylose and reducing the amylose content [23,24]. In addition, the amylose content of red adzuki bean starch decreased significantly with increasing time of amylose removal in warm water, likely because the lipid removal from the starch granules before warm water treatment reduced the binding of lipids to amylose, making it easier for amylose to move out [25]. On the other hand, it might be because water entered the starch granules more efficiently at a certain temperature, which swelled the starch granules and reduced the intermolecular forces [26], and with the increase in treatment time, it destroyed the internal structure of the red adzuki bean starch granules and promoted the movement of amylose.

The amylose content of the samples obtained after the combined treatment (DA-HCl) decreased further, i.e., hydrochloric acid modification after DA exacerbated the decreasing amylose content. The amylose content decreased significantly with increasing HCl concentration or warm water deamylose treatment time. In addition, the amylose content of red adzuki bean starch decreased with increasing hydrochloric acid addition at the same deamylose time, and the amylose content of the samples decreased gradually with increasing deamylose treatment time at the same hydrochloric acid concentration. This might be because hydrochloric acid mainly affected the amorphous region of the red adzuki bean starch, which was acidolyzed into small short-chain molecules that penetrated the starch granules [27]. 

It was also possible that the treatment condition of warm water destroyed the amorphous zone of the starch granules, making it easier for the hydrochloric acid to enter the granules and reduce the amylose content. There was no significant difference in the amylose content of samples DA2.5-H0.6 and DA2.5-H0.9. After the water bath deamylose treatment, HCl may have entered the red adzuki bean starch granules more easily and caused damage to the amorphous zone. The high concentration of hydrochloric acid further affected the crystalline zone and the branched starch was cleaved, resulting in rearrangements such as amylopectin–amylose and amylose–amylose [5], ultimately leading to insignificant changes in amylose content.

### 4.6. Pasting Property Evaluation

The peak, trough, and final viscosities decreased gradually with the increase in the treatment time. In contrast, the attenuation value, regeneration value, gelatinization temperature and pasting time all increased gradually. This indicates that the maximum swelling power of the red adzuki bean starch granules decreased after the water bath deamylose modification, a result reflected in Table 3, which may be due to the reduction in the size of the modified starch granules [28]. The thermal stability of the red adzuki bean starch granules increased after pasting. Nevertheless, the cold paste stability decreased and was prone to regrowth, possibly due to reduced amylose content. In contrast, the increase in pasting temperature and pasting time indicated that the modified starch granules were more challenging to paste, which may be related to the reduced size of the modified red adzuki bean starch granules [29] and the more stable crystalline structure of the starch granules [30].

After HCl treatment, the peak viscosity, final viscosity and trough viscosity of red adzuki starch decreased and were negatively correlated with the concentration of hydrochloric acid added. At the same time, the changes in pasting time and gelatinization temperature were not significant. This indicated that acid digestion destroyed the structure of the amorphous region of the starch and rearranged it to form a more stable crystalline structure [8], resulting in higher crystallinity and lower PV values. On the other hand, acid digestion of red adzuki bean starch by hydrochloric acid enhanced the intermolecular chain forces and limited the swelling of starch granules [31]. In addition, the DA-HCl treatment presented similar effects to the single treatment. Furthermore, the DA-HCl treatment resulted in a more stable structure of red adzuki bean starch granules, with a remarkably higher pasting temperature and pasting time than the native and single-modified starch.

### 4.7. Solubility and Swelling Power Analysis

From the solubility results in Table 3, it can be seen that the solubility of the starch treated with warm water was not significantly different from that of the native starch at 50 °C. However, as the temperature increased, the solubility of the DA starch was considerably lower than that of the native starch. This may be because the amorphous region was destroyed by removing the small molecular weight of amylose by warm water and forming a stable crystalline region; hence, there was no significant difference between samples with the same temperature gradient. In contrast, the solubility of the samples increased slightly with increasing hydrochloric acid concentration at 50–70 °C. This could be related to the destruction of the structure of the amorphous zone of the red adzuki bean starch granules by hydrochloric acid. Water molecules penetrated more easily into the granules and reduced the hydrogen bonding forces between the molecular chains, resulting in a slight increase in solubility [5]. When the temperature was further increased and the pasting temperature of red adzuki bean starch was reached, the starch granules were completely broken, and the solubility increased.

Under low ambient temperature conditions, the swelling power of the hydrochloric-acid-modified red adzuki bean starch granules increased and then decreased without significant difference. This might be related to the rupture of the amorphous region of the red adzuki bean starch granules after the low concentration of hydrochloric acid treatment and the simple rearrangement between molecules. The rearranged starch molecules were unstable, and hydrogen bonding between adjacent starch polymers was disrupted [32], resulting in the low pasting temperature and high swelling power of the samples. As the concentration of hydrochloric acid increased, the amorphous zone lamellae underwent acid digestion and formed a more stable crystalline structure or penetrated the starch granules, resulting in a higher pasting temperature, more excellent stability and lower swelling power. 

In addition, acid digestion destroyed the crystalline zone, reducing swelling power [33]. The swelling power increased when the treatment temperature was higher than the starch granule pasting temperature, but all were lower than that of native red adzuki bean starch. The swelling power of the warm-water-modified samples was remarkably lower than that of the native starch. Structural changes within the starch granules following the deregulation of the amylose in a warm water bath may be responsible for the reduced swelling power. The change in crystallinity has been attributed to the growth of crystals or the refinement of already existing crystals [32]. The swelling power pattern of the DA-HCl-treated samples was insignificant, but all were lower than that of the native starch, which was also consistent with the results of peak viscosity.

### 4.8. Principal Component Analysis

The correlation between multiscale structures and physicochemical properties among native starch, DA starch, HCl-modified starch and dual-modified starch can be verified using PCA scoring plots (Figure 3A) and loading plots (Figure 3B). The accumulated contribution of PC1 (57.0%) and PC2 (17.2%) was 74.2%. The native red adzuki bean starch was located in the fourth quadrant and the hydrochloric-acid-treated samples gradually moved toward the third quadrant as the sample concentration increased. The samples treated with deamylose were located in the first quadrant and moved toward the second quadrant with increasing treatment time. The composite-treated samples were mainly distributed in the second and third quadrants. The combination of loading plots and correlation plots further reflects the structure–physicochemical property linkage of starch samples. The values for pasting characteristics (SB, FV, TV, PV), RC and amylose content were located on the positive half-axis of PC1 and presented a highly significant positive correlation with the short A chain. Solubility was located on the negative half-axis of PC1. It showed a highly significant negative correlation analysis with the A chain and a positive correlation with the B3 chain. This demonstrated that the structural distribution of the short A and long B3 chains greatly influenced the physicochemical properties of starch.

### 4.9. Mechanism of EBI Pretreatment Modification

As shown in Figure 4, the intrinsic structure of starch granules is complex, and differences in the structural characteristics of these starches could often determine their functional properties. In conjunction with the composition of the starch structure, we must explore the effect of removing amylose on the modification of starch granules (Figure 4). Acid hydrolysis is an attractive modification method in the starch industry as it can effectively alter the internal structure and functional properties of starch without destroying the morphology of the starch granules [5]. As can be seen in Figure 4, the acid penetrated the starch granules, preferentially attacking the granule surface, then eroded the amorphous regions of the starch granules and degraded the starch molecules [34]. Preferential hydrolysis of the amorphous region increased relative crystallinity and short-range ordering, as shown in Table 1.

In addition, Chen et al. [22] also compared the acid hydrolysis mode of waxy starch and G80 starch and found that waxy starch with more pores on the surface of the starch granules could be hydrolyzed from the interior to the periphery of the starch, and obtained the conclusion that the acid hydrolysis efficiency might be related to the porosity of the starch granules. In our research, the native red adzuki bean starch granules were primarily smooth, oval or kidney-shaped, with a few having raised surfaces (Figure 1). In contrast, the surface of the starch granules was concave after extraction of amylose with warm water, and the percentage of deformation increased with increasing deamylose time. This also explained the higher degree of corrosion damage to red adzuki bean starch from deamylose–acid hydrolysis.

Starch is divided into crystalline and amorphous regions. To our knowledge, the deamylose treatment causes the amylose with a smaller molecular weight to penetrate the starch surface. With the entry of water molecules, the glycosidic bonds between the starch molecules were broken (Figure 4). At the same time, with the increase in deamylose time, the structure of the crystalline region was also disrupted. The starch molecular structure was rearranged, resulting in a decrease in the A and B1 chains and an increase in the proportion of B2 and B3 chains (Table 1). In addition, acid hydrolysis also led to amylose and amylopectin degradation. A hypothesis of cleavage of the a-(1/6) branching point through acid hydrolysis has been reported [33]. It is known that acids preferentially hydrolyze the amorphous fraction of starch granules, resulting in a higher percentage of crystallinity (Table 1). The increased crystallinity of acid-hydrolyzed starch samples might lead to granule resistance to swelling, which may be caused by the enhanced bonding within the granules [17]. Furthermore, hydrogen bonds between adjacent starch polymers were broken during acid treatment and eroded amorphous regions, reducing swelling forces [32]. The amylose leached from the deamylose–acid-hydrolyzed samples was partially dissolved in the granules and was susceptible to rapid hydrolysis [35]. Moreover, disrupting the molecular structure of the amorphous regions of starch leads to the polymerization of starch molecules and decreased pasting viscosity [8].

## 5. Conclusions

In this research, one facility was proposed for modifying red adzuki bean starch through warm water deamylose treatment, and its feasibility for improving starch properties was verified by exploring its multiscale structure. The effect of hydrochloric acid digestion on the deamylose was further investigated. The surface of red adzuki bean starch granules was deformed and depressed after deamylose; the AC and A chain content were significantly reduced, while the ratio of 1047 cm^−1^/1022 cm^−1^ and RC were first increased and then reduced. It was confirmed that the small molecular weight of amylose was mainly enriched on the surface of the starch granules, and the short-time deamylose treatment made the starch structure more compact, while the extended time caused damage to the granule structure. After acid digestion, the starch granules showed wrinkles and a significant decrease in PV. In addition, the A chain proportion and the AC decreased, indicating that hydrochloric acid mainly affected the structure of the amorphous region of the starch granules. DA-HCl treatment revealed a more significant decrease in AC, RC and A chain content and a substantial increase in solubility. This study shows that the amylose content severely affects the multiscale structure and physicochemical properties of starch, which also provides a basis for developing degradable materials and food additives.

## Figures and Tables

**Figure 1 foods-12-03366-f001:**
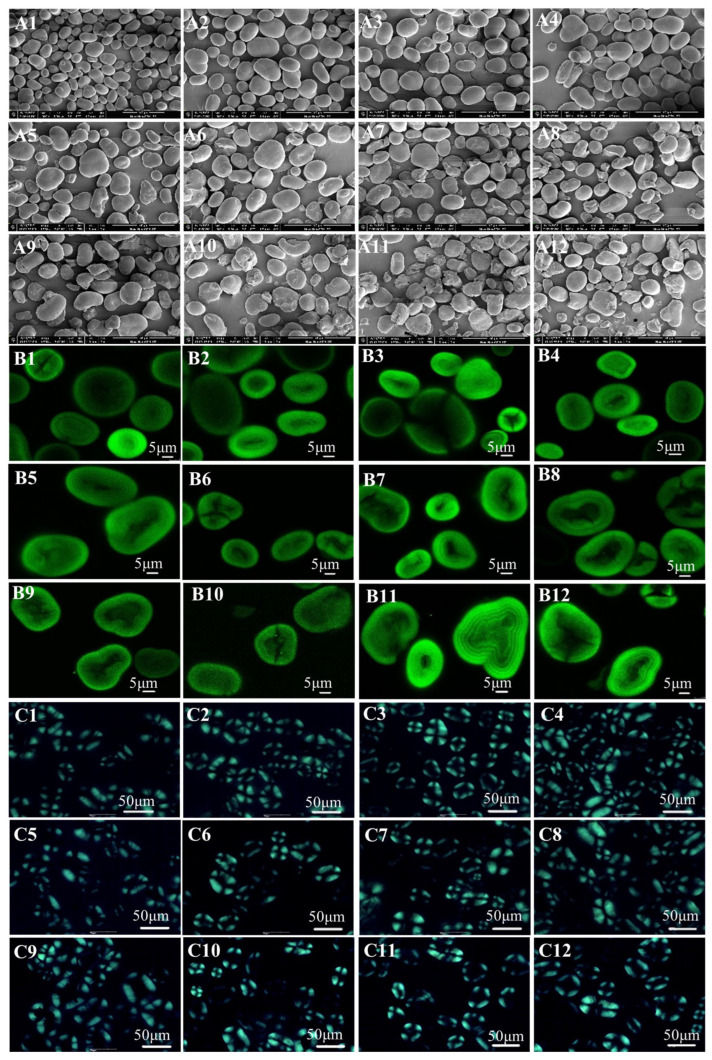
Scanning electron microscope (**A**), laser confocal microscope (**B**) and polarizing microscope (**C**) images of the native and modified red adzuki starch. 1, Native; 2, H-0.3; 3, H-0.6; 4, H-0.9; 5, DA-1; 6, DA1-H0.3; 7, DA1-H0.6; 8, DA1-H0.9; 9, DA-2.5; 10, DA2.5-H0.3; 11, DA2.5-H0.6; 12, DA2.5-H0.9.

**Figure 2 foods-12-03366-f002:**
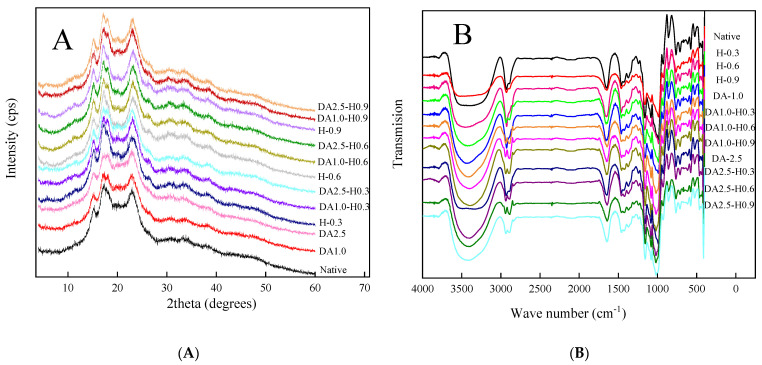
X-ray diffraction (**A**) and FT-IR spectra (**B**) of the native and de-amylose-, hydrochloric-acid- and dual-modified red adzuki starch.

**Figure 3 foods-12-03366-f003:**
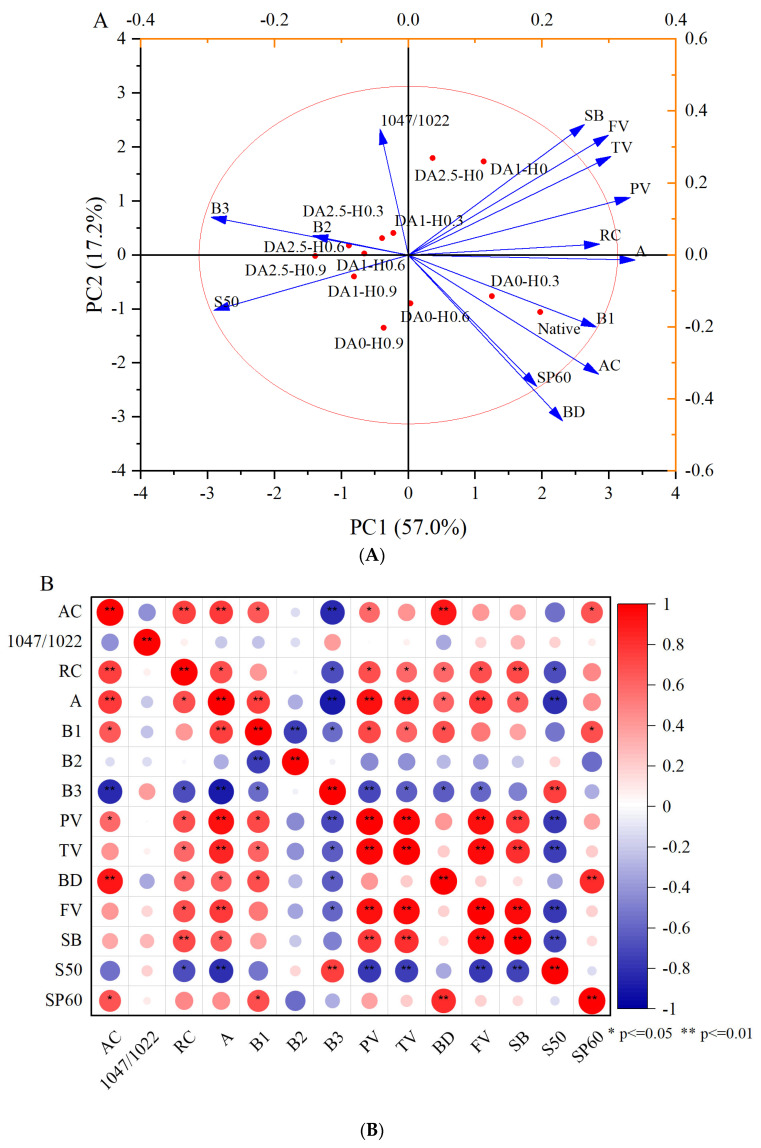
Score plot and loading plot (**A**) and Pearson’s correlation analysis (**B**) of the native and de-amylose-, hydrochloric-acid- and dual-modified red adzuki starch.

**Figure 4 foods-12-03366-f004:**
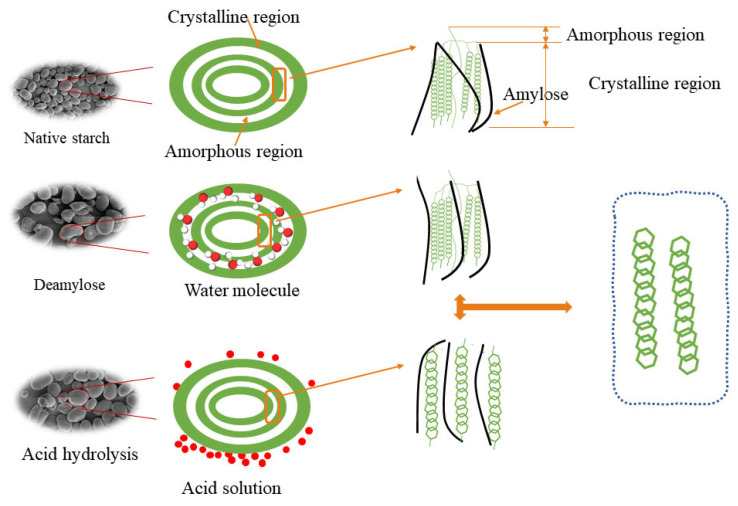
The proposed mechanism of acid-hydrolysis-, deamylose- and dual-modified red adzuki starch.

**Table 1 foods-12-03366-t001:** The amylose content, relative crystallinity and FT-IR ratio of the native, and de-amylose-, hydrochloric-acid- and dual-modified red adzuki bean starch ^1, 2^.

Samples		AC (%)	1047 cm^−1^/1022 cm^−1^	RC (%)	Chain Length Distributions (%)			
DA	HCl				DP6–12 (A)	DP13–24 (B_1_)	DP25–36 (B_2_)	DP > 36 (B_3_)
0	0	29.74 ± 0.07 ^a^	1.025 ± 0.000 ^f^	25.53 ± 0.19 ^c^	44.85 ± 0.93 ^a^	39.17 ± 0.94 ^a^	14.73 ± 0.61 ^f^	1.26 ± 0.62 ^g^
	0.3	29.27 ± 0.03 ^b^	1.049 ± 0.01 ^bcd^	27.28 ± 0.26 ^a^	42.36 ± 0.51 ^bc^	37.25 ± 0.35 ^b^	16.26 ± 0.36 ^de^	4.14 ± 1.22 ^ef^
	0.6	28.63 ± 0.03 ^c^	1.043 ± 0.002 ^cde^	26.09 ± 0.17 ^b^	41.46 ± 0.20 ^cd^	33.71 ± 0.4 ^de^	19.96 ± 1.35 ^b^	4.88 ± 0.74 ^de^
	0.9	28.21 ± 0.00 ^d^	1.025 ± 0.002 ^f^	24.61 ± 0.15 ^d^	40.30 ± 0.56 ^de^	34.89 ± 0.54 ^cd^	18.52 ± 0.68 ^bc^	6.30 ± 0.41 ^cd^
1	0	27.64 ± 0.03 ^e^	1.046 ± 0.003 ^bcd^	26.23 ± 0.23 ^b^	42.90 ± 0.15 ^bcde^	35.57 ± 0.66 ^c^	18.84 ± 0.48 ^bc^	2.70 ± 0.99 ^fg^
	0.3	27.27 ± 0.07 ^f^	1.041 ± 0.004 ^de^	25.37 ± 0.20 ^c^	41.54 ± 0.66 ^cd^	33.00 ± 1.28 ^e^	21.69 ± 0.97 ^a^	3.79 ± 0.97 ^ef^
	0.6	26.80 ± 0.03 ^g^	1.037 ± 0.002 ^e^	24.08 ± 0.16 ^e^	40.16 ± 0.63 ^ef^	34.15 ± 0.21 ^cde^	19.00 ± 0.01 ^bc^	6.70 ± 0.42 ^bc^
	0.9	26.46 ± 0.10 ^h^	1.021 ± 0.008 ^f^	22.20 ± 0.21 ^f^	40.40 ± 0.71 ^de^	34.43 ± 0.89 ^cde^	18.44 ± 0.79 ^bc^	6.73 ± 0.61 ^bc^
2.5	0	26.39 ± 0.07 ^h^	1.051 ± 0.011 ^abc^	25.54 ± 0.22 ^c^	41.40 ± 0.56 ^cde^	34.19 ± 0.27 ^cde^	16.55 ± 0.64 ^de^	7.87 ± 0.19 ^abc^
	0.3	26.01 ± 0.10 ^i^	1.058 ± 0.004 ^a^	24.37 ± 0.21 ^de^	40.45 ± 0.35 ^de^	35.48 ± 0.67 ^c^	15.83 ± 0.24 ^ef^	8.25 ± 0.78 ^ab^
	0.6	25.67 ± 0.03 ^j^	1.053 ± 0.006 ^ab^	22.12 ± 014 ^f^	40.42 ± 0.17 ^de^	34.20 ± 0.57 ^cde^	17.53 ± 0.11 ^cd^	7.85 ± 0.49 ^abc^
	0.9	25.61 ± 0.03 ^j^	1.049 ± 0.001 ^bcd^	21.57 ± 0.06 ^g^	39.06 ± 0.36 ^f^	33.83 ± 0.24 ^de^	17.72 ± 0.40 ^cd^	9.40 ± 0.28 ^a^

^1^ DA, deamylose; AC, amylose content; RC, relative crystallinity. ^2^ Data are expressed as mean ± standard deviation. Means with different letters within the same column are significantly different (*p* < 0.05).

**Table 2 foods-12-03366-t002:** Pasting parameters of the native and de-amylose-, hydrochloric-acid- and dual-modified red adzuki bean starch ^1,2^.

DA	HCl/	PV (cP)	TV (cP)	BD (cP)	FV (cP)	SB (cP)	PT (min)	GT (°C)
0	0	4510 ± 76 ^a^	3798 ± 42 ^a^	712 ± 34 ^b^	5094 ± 122 ^c^	1296 ± 81 ^d^	4.57 ± 0.05 ^e^	73.93 ± 0.60 ^fg^
	0.3	2599 ± 2 ^d^	1836 ± 15 ^d^	763 ± 13 ^a^	4169 ± 5 ^d^	2333 ± 20 ^c^	4.44 ± 0.05 ^ef^	73.45 ± 0.00 ^g^
	0.6	1552 ± 40 ^f^	1045 ± 32 ^f^	507 ± 8 ^c^	1725 ± 42 ^g^	681 ± 11 ^f^	4.30 ± 0.04 ^f^	74.70 ± 0.57 ^ef^
	0.9	1049 ± 8 ^g^	618 ± 2 ^h^	431 ± 10 ^d^	1003 ± 8 ^i^	386 ± 11 ^g^	4.27 ± 0.00 ^f^	75.10 ± 0.07 ^e^
1	0	3514 ± 7 ^b^	3426 ± 11 ^b^	88 ± 18 ^gh^	6705 ± 34 ^a^	3279 ± 45 ^a^	5.63 ± 0.14 ^b^	77.90 ± 0.56 ^d^
	0.3	1791 ± 11 ^e^	1523 ± 71 ^e^	203 ± 10 ^e^	2393 ± 115 ^e^	870 ± 44 ^e^	5.27 ± 0.00 ^cd^	78.30 ± 0.00 ^d^
	0.6	1043 ± 13 ^g^	908 ± 1 ^g^	135 ± 12 ^f^	1265 ± 8 ^h^	357 ± 7 ^g^	5.20 ± 0.00 ^d^	78.65 ± 0.56 ^cd^
	0.9	719 ± 1 ^i^	605 ± 4 ^h^	114 ± 4 ^fg^	810 ± 2 ^j^	205 ± 6 ^h^	5.14 ± 0.09 ^d^	78.68 ± 0.53 ^cd^
2.5	0	3151 ± 17 ^c^	3121 ± 16 ^c^	30 ± 1 ^i^	5614 ± 120 ^b^	2493 ± 104 ^b^	6.44 ± 0.33 ^a^	79.53 ± 0.60 ^bc^
	0.3	1598 ± 16 ^f^	1480 ± 10 ^e^	118 ± 6 ^fg^	2177 ± 13 ^f^	697 ± 23 ^f^	5.67 ± 0.00 ^b^	80.30 ± 0.49 ^ab^
	0.6	969 ± 14 ^h^	889 ± 16 ^g^	81 ± 2 ^h^	1190 ± 18 ^h^	302 ± 2 ^g^	5.57 ± 0.05 ^b^	81.15 ± 0.64 ^a^
	0.9	700 ± 9 ^i^	632 ± 5 ^h^	68 ± 4 ^h^	829 ± 12 ^j^	197 ± 7 ^h^	5.50 ± 0.04 ^bc^	80.73 ± 0.04 ^a^

^1^ DA, deamylose; PV, peak viscosity; TV, trough viscosity; BD, breakdown; FV, final viscosity; SB, setback; PT, peak time; GT, gelatinization temperature. ^2^ Data are expressed as mean ± standard deviation. Means with different letters within the same column are significantly different (*p* < 0.05).

**Table 3 foods-12-03366-t003:** Solubility and swelling power of the native and de-amylose-, hydrochloric-acid- and dual-modified red adzuki bean starch ^1,2^.

Treatment		Solubility (%)					Swelling Power (g/g)				
DA	HCl	50 °C	60 °C	70 °C	80 °C	90 °C	50 °C	60 °C	70 °C	80 °C	90 °C
0	0	0.57 ± 0.23 ^de^	2.40 ± 0.08 ^c^	6.96 ± 0.66 ^de^	8.55 ± 0.17 ^g^	8.48 ± 0.32 ^h^	2.41 ± 0.04 ^d^	4.44 ± 0.06 ^ab^	9.73 ± 0.54 ^ab^	14.80 ± 0.08 ^a^	23.43 ± 0.24 ^a^
	0.3	0.62 ± 0.08 ^cde^	3.37 ± 0.49 ^b^	7.82 ± 0.63 ^cd^	6.41 ± 0.49 ^j^	4.25 ± 0.28 ^j^	2.41 ± 0.07 ^d^	4.63 ± 0.18 ^a^	10.29 ± 0.11 ^a^	14.21 ± 0.28 ^b^	14.61 ± 0.80 ^e^
	0.6	0.93 ± 0.09 ^bcd^	3.77 ± 0.48 ^b^	7.05 ± 0.03 ^de^	10.09 ± 0.17 ^f^	12.03 ± 0.47 ^f^	2.26 ± 0.28 ^d^	4.29 ± 0.13 ^bc^	9.12 ± 0.75 ^bc^	12.28 ± 0.00 ^c^	15.72 ± 0.11 ^e^
	0.9	1.07 ± 0.27 ^b^	4.85 ± 0.10 ^a^	8.06 ± 0.69 ^bcd^	7.95 ± 0.46 ^gh^	21.79 ± 0.37 ^b^	2.18 ± 0.42 ^d^	4.08 ± 0.09 ^cd^	8.73 ± 0.55 ^cd^	10.54 ± 0.72 ^d^	17.89 ± 0.59 ^d^
1	0	0.51 ± 0.06 ^e^	0.52 ± 0.04 ^f^	7.10 ± 0.01 ^de^	7.45 ± 0.75 ^hi^	3.78 ± 0.21 ^j^	3.31 ± 0.23 ^bc^	3.62 ± 0.16 ^efg^	8.23 ± 0.59 ^cde^	10.52 ± 0.16 ^d^	19.73 ± 0.42 ^c^
	0.3	0.84 ± 0.30 ^bcde^	1.34 ± 0.17 ^e^	7.36 ± 0.28 ^d^	13.00 ± 0.12 ^d^	9.72 ± 0.23 ^g^	3.41 ± 0.26 ^abc^	3.42 ± 0.07 ^g^	8.32 ± 0.25 ^cde^	10.36 ± 0.23 ^de^	20.09 ± 0.35 ^c^
	0.6	0.82 ± 0.07 ^bcde^	1.29 ± 0.04 ^e^	9.00 ± 0.42 ^ab^	12.05 ± 0.23 ^e^	13.27 ± 0.56 ^e^	3.56 ± 0.14 ^ab^	3.44 ± 0.02 ^fg^	7.88 ± 0.49 ^de^	9.43 ± 0.06 ^fg^	22.58 ± 0.99 ^ab^
	0.9	0.80 ± 0.07 ^bcde^	1.62 ± 0.28 ^de^	9.23 ± 0.21 ^a^	14.35 ± 0.16 ^bc^	17.56 ± 0.67 ^c^	3.11 ± 0.06 ^c^	3.45 ± 0.31 ^fg^	7.61 ± 0.43 ^e^	8.83 ± 0.10 ^h^	15.06 ± 0.34 ^e^
2.5	0	0.71 ± 0.15 ^bcde^	0.54 ± 0.08 ^f^	6.17 ± 0.66 ^e^	6.93 ± 0.12 ^ij^	6.51 ± 0.36 ^i^	3.63 ± 0.14 ^a^	3.70 ± 0.11 ^efg^	8.59 ± 0.37 ^cde^	10.26 ± 0.06 ^de^	21.65 ± 0.24 ^b^
	0.3	0.99 ± 0.00 ^bc^	1.29 ± 0.01 ^e^	8.70 ± 0.19 ^abc^	13.99 ± 0.27 ^c^	8.02 ± 0.40 ^h^	3.49 ± 0.13 ^ab^	4.08 ± 0.21 ^cd^	7.80 ± 0.14 ^de^	9.86 ± 0.01 ^ef^	15.67 ± 0.23 ^e^
	0.6	1.00 ± 0.22 ^bc^	2.18 ± 0.57 ^cd^	8.98 ± 0.05 ^ab^	14.92 ± 0.05 ^b^	15.97 ± 0.53 ^d^	3.36 ± 0.05 ^abc^	3.89 ± 0.04 ^de^	8.03 ± 0.19 ^de^	9.61 ± 0.03 ^fg^	18.34 ± 0.55 ^d^
	0.9	1.43 ± 0.06 ^a^	2.31 ± 0.27 ^c^	9.38 ± 0.70 ^a^	16.62 ± 0.02 ^a^	29.43 ± 0.10 ^a^	3.41 ± 0.01 ^abc^	3.77 ± 0.06 ^def^	7.65 ± 0.09 ^e^	9.20 ± 0.02 ^gh^	15.48 ± 0.30 ^e^

^1^ DA, deamylose. ^2^ Data are expressed as mean ± standard deviation. Means with different letters within the same column are significantly different (*p* < 0.05).

## Data Availability

The data supporting this study are available on request from the corresponding author.

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
