# Peer review of "Spotlight on the Multiscale Structural and Physicochemical Properties of Red Adzuki Bean Starch through Partial Amylose Removal Combined with Hydrochloric Acid"

_foods, 2023, doi:10.3390/foods12183366_

Round 1

Reviewer 1 Report

In lines 80 and 90 the concentration is mentioned first, between parentheses, and then the name of the reagent used (methanol, NaOH). However, later in the text (lines 105,106, 107 and 111), the reagent is named first and then the concentration; it is suggested to standardize the writing using the last format described here

It is suggested to reorganize the information, placing in each epigraph only the images, results and discussion of each experiment separately; Since the way in which the information is presented, you have to go back in the document to understand what is being discussed.

 Figure 2: It is correct to represent the IR spectra and the diffractograms of each sample, respectively, on the same graph. However, since the objective is the comparison of results, it is recommended to use colors that make it possible to distinguish one sample from another and appreciate the changes in case they exist.

On the other hand, in the diffractogram, from the way in which the abscissa axis is named, it is not clear what the unit of measure is. It is suggested to change to 2theta (degrees)

Minor editing of English language required

Author Response

1. In lines 80 and 90 the concentration is mentioned first, between parentheses, and then the name of the reagent used (methanol, NaOH). However, later in the text (lines 105,106, 107 and 111), the reagent is named first and then the concentration; it is suggested to standardize the writing using the last format described here.

Response: We do appreciate you for giving us positive and insightful comments. As you requested, we have corrected the formatting and checked the full text for consistency.

2. It is suggested to reorganize the information, placing in each epigraph only the images, results and discussion of each experiment separately; Since the way in which the information is presented, you have to go back in the document to understand what is being discussed.

Response: We do appreciate you for giving us positive and insightful comments. We apologize for any distress caused to you. We have reorganized the information as you suggested.

3. Figure 2: It is correct to represent the IR spectra and the diffractograms of each sample, respectively, on the same graph. However, since the objective is the comparison of results, it is recommended to use colors that make it possible to distinguish one sample from another and appreciate the changes in case they exist.

Response: Thanks for your positive suggestion. We strongly agree with your suggestion. We have distinguished the curves in Figure 2 with different colors.

4. On the other hand, in the diffractogram, from the way in which the abscissa axis is named, it is not clear what the unit of measure is. It is suggested to change to 2theta (degrees).

Response: We apologize for the trouble we caused you and have corrected the abscissa axis of the diffractogram as you suggested.

5. Minor editing of English language required.

Response: Thanks for your positive suggestion. The whole manuscript has been revised for English usage by a scientific and language editing expert.

Reviewer 2 Report

 The authors have investigated the multi-level structural and physicochemical properties of red adzuki bean starch with a high amylose content by using a deliquescence treatment and different concentrations of hydrochloric acid. Also, the effect of acid hydrolysis on the "structure-property" relationship by first removing the amylose were also elucidated.

This work adds to knowledge in starch science research and industrial applications for the acid processing of starch-based foods, particularly about the most important factors controlling the structure and function of starch.

The abstract is written correctly.

The introduction describes the problem quite well. Well-chosen literature is cited.

In the material and methods section, some methods need to be briefly described. In the 2.2. Starch separation subsection, 2.10. Amylose content subsection, please briefly described the method used.

The results are correctly described, the tables and figures are easy to read.

Discussion of results are correctly written. The literature is properly selected.

The references sections need to be revised and prepared according to the Instructions for Authors.

 Manuscript can be further improved taking following points into consideration.

Lines 83, 84, 91, 121: … what is 4000 r/min?

Line 110: 2.6.3. Confocal laser scanning microscopy (PLM) – please verify. The same subtitle is at 2.6.2.

Line 128: … was collected …. Instead of …. was reacted …. What is the resolution and the number of scans, please add!

Line 137: What is the device used for pasting properties measurement?

Line 143: How was determined solubility and swelling power?

3.1. Morphological characteristics

Please add in the text, see Figure 1.

Observation of the microscopic images of red adzuki bean starch (Figure 1) …..

 Line 195: Solubility and Swelling Power instead of Solubility and SP

Line 208: … Figure 1. instead of … Figure. 1.

Line 227: … (Figure 1B1) instead of … (Figure. 1B1)

Line 232: … Figure 1C1-C12 instead of … Figure. 1C1-C12

Line 289: Figure 2A … instead of Figure. 2A ….

Line 307: …. in the wavenumber range of 400-4000 cm-1 …. instead of … in the 4000-400 cm-1 range

Line 393: Solubility and Swelling Power analysis instead of Solubility and SP analysis

Line 435:  4.8. Principal component analysis instead of 4.8. PCA analysis

Line 441: …. the PC1 instead of …. the X-axis

Line 533: please cite the Figure 4 in the text.

Minor edits needed; should be read by an Editor in charge of English editing.

Author Response

Minor edits needed; should be read by an Editor in charge of English editing. The authors have investigated the multi-level structural and physicochemical properties of red adzuki bean starch with a high amylose content by using a deliquescence treatment and different concentrations of hydrochloric acid. Also, the effect of acid hydrolysis on the "structure-property" relationship by first removing the amylose were also elucidated. This work adds to knowledge in starch science research and industrial applications for the acid processing of starch-based foods, particularly about the most important factors controlling the structure and function of starch. The abstract is written correctly. The introduction describes the problem quite well. Well-chosen literature is cited. In the material and methods section, some methods need to be briefly described. In the 2.2. Starch separation subsection, 2.10. Amylose content subsection, please briefly described the method used. The results are correctly described, the tables and figures are easy to read. Discussion of results are correctly written. The literature is properly selected. The references sections need to be revised and prepared according to the Instructions for Authors.

Response: We greatly appreciate your time, patience, and helpful comments regarding our manuscript. The whole manuscript has been revised for English usage by a scientific and language editing expert. The language of the present version has significantly improved. Moreover, the methodology is briefly described in sections 2.2 and 2.10. The references have been corrected according to the Instructions for Authors.

Manuscript can be further improved taking following points into consideration.

1. Lines 83, 84, 91, 121: … what is 4000 r/min?

Response: We are very apologetic for the trouble we have caused you. What we were trying to convey was the stirring rate, i.e., 4000 rpm. We have checked the full manuscript and made corrections.

2. Line 110: 2.6.3. Confocal laser scanning microscopy (PLM) – please verify. The same subtitle is at 2.6.2.

Response: We apologize for the oversight. We have verified it and corrected it.

3. Line 128: … was collected …. Instead of …. was reacted …. What is the resolution and the number of scans, please add!

Response: Thanks for your suggestion. We have corrected the expression and added the resolution and the number of scans.

4. Line 137: What is the device used for pasting properties measurement?

Response: Thanks for you pointing this out. The device used for pasting properties measurement is Rapid Visco Analyzer (RVA). We have made additions to the manuscript.

5. Line 143: How was determined solubility and swelling power?

Response: Thanks for you pointing this out. We have added the determination of solubility and swelling power in 2.12. Measurement of solubility and swelling power subsection.

6. 3.1. Morphological characteristics Please add in the text, see Figure 1.

Observation of the microscopic images of red adzuki bean starch (Figure 1) …..

Response: Thanks for your positive suggestion. We have added to 3.1. Morphological characteristics, as you suggested.

7. Line 195: Solubility and Swelling Power instead of Solubility and SP. Line 208: … Figure 1. instead of … Figure. 1. Line 227: … (Figure 1B1) instead of … (Figure. 1B1). Line 232: … Figure 1C1-C12 instead of … Figure. 1C1-C12. Line 289: Figure 2A … instead of Figure. 2A ….

Response: Thank you very much for pointing this out. We have corrected and checked the full text.

8. Line 307: …. in the wavenumber range of 400-4000 cm-1 …. instead of … in the 4000-400 cm-1

Response: Thank you very much for pointing this out. We have corrected it according to your suggestion.

9. Line 393: Solubility and Swelling Power analysis instead of Solubility and SP analysis.

Response: Thank you very much for pointing this out. We have corrected it according to your suggestion.

10. Line 435: 8. Principal component analysis instead of 4.8. PCA analysis.

Response: Thanks for pointing this out. We have corrected it according to your suggestion.

11. Line 441: …. the PC1 instead of …. the X-axis.

Response: Thanks for pointing this out. We have corrected it according to your suggestion.

12. Line 533: please cite the Figure 4 in the text.

Response: Thanks for your positive suggestion. We have cited Figure 4 in the manuscript, as you suggested.